# 3D Modelling for Photonic Crystal Structure in *Papilio maackii* Wing Scales

**DOI:** 10.3390/ma15093334

**Published:** 2022-05-06

**Authors:** Shu Yang, Yingwen Wang, Weihong Gao

**Affiliations:** School of Textiles and Fashion, Shanghai University of Engineering and Science, Shanghai 201620, China; wyw1975193932@163.com (Y.W.); gaoweihong@sues.edu.cn (W.G.)

**Keywords:** *Papilio maackii*, wing scales, photonic crystal, 3D modelling, band gap

## Abstract

As a typical representative of natural structural colors, the wings of butterflies living in different zones present colors due to different chromogenic mechanisms. In this work, *Papilio maackii*, a common species of butterfly living in China, was studied in order to clarify the photophysics of its wing scales. A FESEM was applied to observe the microstructure of the scales, and we found that they have a periodic photonic crystal structure. X-ray photoelectron spectroscopy was applied to clarify the wings’ chemical composition. Additionally, the optical properties of the scales were investigated using a UV-vis-NIR microspectrophotometer. Then, a simplified three-dimensional photonic crystal model was built according to the microstructure of the wing scales, and the plane-wave expansion method was used to calculate the band gap. The correlation between the calculated band gap and the practical reflective spectrum was also established for the wing scales of *Papilio maackii*.

## 1. Introduction

The colors of butterfly wings usually originate from chemical pigments, physical structure, or a mixture of both [1,2,3]. Color of a purely physical origin is called ‘structural color’ or ‘iridescence’, which is often angle-dependent and has been of interest to researchers over the past two decades [4,5,6]. Since the 1950s, studies investigating mechanisms of structural color have been increasing gradually, along with developments in electron microscopic technology [7]. Structural color is considered to be caused by three mechanisms: thin-film reflectors, diffraction gratings, or scattering of light waves [2]. Some, however, do not fall into the above categories, such as photonic crystals, first verified by Parker in 2001 [8].

The specific mechanisms of photonic crystals in terms of the structural color of butterfly wing scales are still obscure and controversial [9,10,11]. One possible reason for this is biological diversity, and another is the limitations of optical detection in that the movements of light on a micro–nano scale cannot be observed directly. Additionally, and interestingly, many creatures use all or several of these mechanisms together to skillfully show off a complicated color [12].

Butterflies living in different climate zones have diverse colors, on account of hiding from the surrounding environment and protecting their bodies from cold or hot climates via efficient sunlight absorption of their wings’ surface scales [13]. Therefore, it is necessary to conduct further research on more species of butterflies living in diverse zones.

*Papilio maackii* is a butterfly species living in the temperate zone and can usually be found in central Asia, Japan, and China. There have been some previous studies on the structure and optical properties of *Papilio maackii* [14,15,16,17].

In this paper, the structural characteristics and chemical composition of *Papilio maackii* wing scales were investigated; then, the complicated microstructures were simplified into 3D photonic crystal models, which can be used in the band gap calculation through the plane-wave expansion method. Finally, theoretical band gaps of photonic crystals were calculated using the plane-wave expansion method, and the relationship between the theoretical band gap and the optical spectrum results was analyzed. As far as we know, this is the first time such results have been reported publicly.

## 2. Materials and Methods

### 2.1. Materials

Specimen of *Papilio maackii* were acquired from the Honglin Teaching Instrument Company, and ten specimens were investigated for each experiment. On each butterfly wing, two sections of different colors were selected, named Section 1 and Section 2. Solutions, such as a normal saline solution and ethanol, were bought from Aladdin.

### 2.2. Observation

Specimens were rinsed in a normal saline solution (0.65% NaCl) for 10 min in order to remove impurities. The morphology of the wing scales was characterized by SEM (Hitachi SU-70, Tokyo, Japan) after coating with 20 nm platinum.

### 2.3. XPS

X-ray photoelectron spectroscopy (AXIS ULTRA-DLD) was applied to test the chemical composition of butterfly wings. X-ray photoelectron spectra (XPS) were recorded using monochromated Al Kα radiation (1486.6 eV) with 30 eV pass energy in 0.5 eV steps over an area of 650 × 650 μm. All binding energies were referenced to the C 1 s peak at 285 eV. Before XPS measurement, the specimens were degassed under a high-vacuum condition (<10^−7^ Pa) to remove adsorbed water and oxygen.

### 2.4. Spectrometer

One important characteristic of photonic crystals is their angle dependency. That is, wings will show different colors under incident lights from different directions [18]. Another characteristic is that the color will change with different interstitial substances [19,20]. The wing scale is a binary photonic system made of chitin as its entity and air as its interstitial substance [4,21].

UV-vis-NIR micro spectrophotometers (Ocean USB2000) were applied to characterize the optical property of wing scales. The wavelength range was within 350–950 nm, and the temperature of the laboratory was 20 °C with a relative humidity of 65%.

In order to clarify two characteristics of photonic crystals, two spectrum experiments were applied. In the first test, the beam angle of the incident light was set as 90°, 60°, and 30° to the plane of wing. Additionally, in the second test, ethanol was dropped onto the scales, meaning that the interstitial substance of the photonic crystal was then ethanol instead of air.

### 2.5. Model and Calculation

It is proposed that wing scales are 3D photonic crystals composed of an entity material and interstitial substance. The entity material is chitin [22], whose dielectric constant is 2.43, and the interstitial substance is air, whose dielectric constant is 1. The microstructure of the scales was simplified into calculation models. Then, the plane-wave expansion method was used to calculate the band gap.

## 3. Results

### 3.1. Microstructure of Wing Scales

The structure of Section 1 on wings of *Papilio maackii* is shown in Figure 1. Figure 1a is the image of the whole scale; it can be seen that scales are tile-packed with some overlap. In Figure 1b, a single scale is shown with a narrower root and hackly tip. Figure 1c,d clearly shows the microstructure of a single scale. Ridges run along the longitudinal axis of the scale, and cross ribs connect neighboring ridges transversely. Cross ribs are irregular and sometimes fuse with neighboring cross ribs. In addition, trabeculae exist underneath and are sheet-like. These form hexagon windows with the cross ribs. In addition, there is an additional substructure on the longitudinal ribs of the scales. The SEM images of Section 2 of *Papilio maackii* are shown in Figure 2, and they are similar to Section 1.

These two sections of *Papilio maackii* both have periodic photonic structures, which are hexagon windows formed by cross ribs and trabeculae between ridges. The average sizes of microstructures of *Papilio maackii* are summarized in Table 1. The size of the whole scale of Section 2 is a little larger than Section 1. The spacing between ridges, thickness of ridges, and spacing between cross ribs of Section 2 are smaller, except for the thickness of the cross ribs.

### 3.2. Chemical Composition

The X-ray photoelectron spectroscopy (XPS) spectra (Figure 3) reveal that the compositions of different sections on wings are nearly the same. The butterfly wings are mainly composed of chitin and protein, including the elements C, N, O, and others. It can, thus, be said that the different colors of wings mainly originate due to physical reasons.

### 3.3. Reflective Spectrum of Incident Light from Different Angles

Reflective spectra of incident lights from different angles are shown in Figure 4. As can be seen from Figure 4a, which is the reflective spectrum of Section 1 of *Papilio maackii*, when the incident angle is 90°, the reflection peak is at 530 nm, which means the color of Section 1 appears as nearly green. This is consistent with Figure 1. From Figure 4c, it can be concluded that when the incident angle decreases, the peak of the reflective spectrum will move to a smaller wavelength, which is called blue shift. Figure 4b is the reflective spectrum of Section 2 of *Papilio maackii*. When the incident angle is 90°, the reflection peak is at about 500 nm, so Section 2 appears blue to the naked eye, which is consistent with the results shown in Figure 2. From Figure 4c, it is found that when the incident angle decreases, blue shift also occurs.

The relationship between the peak of the reflective spectrum and the incident angle, which is shown in Figure 4c, reveals that the reflective spectrum results of Section 1 and Section 2 of *Papilio maackii* are both angle-dependent. This suggests that the colors of these two sections originate from photonic crystals.

### 3.4. Reflective Spectrum after Changing Interstitial Substance

Ethanol was dropped onto two sections of *Papilio maackii*, and the changes in the reflective spectrum are shown in Figure 5. The spectra at five timing points were recorded, namely, before dropping and at 30 s, 1 min, 10 min, and 20 min (when ethanol was completely volatilized) after dropping. The angle of incident light was set as 90°.

From Figure 5a, it can be seen that immediately (30 s) after ethanol was dropped onto the scales, the spectrum shifted to the red region, and the reflectivity declined. After 1 min, the scale was soaked completely, the reflective spectrum moved to the red region, and the reflectivity dropped continually. Furthermore, 10 min after dropping, the spectrum blue shifted and reflectivity lifted since ethanol started to volatilize. At 20 min after dropping, ethanol volatilization was complete, and the reflective spectrum returned to the position it was in before dropping.

As seen in Figure 5b, the results of Section 2 of *Papilio maackii* are very similar to those of Section 1.

Figure 5c shows the relationship between the time after dropping and the peak of the reflective spectrum. After ethanol was dropped into the wing scales, the peak of the reflective spectrum moved to a higher wavelength. When the ethanol volatilized, the peak returned to the original position.

These results verify that the color of the scales can be changed by using a different interstitial substance since the dielectric constant of ethanol is different from air [23]. Therefore, the structural parameters of photonic crystals composed of chitin and ethanol have been changed compared with the original ones composed of chitin and air. When the incident light hits the wing scales, most of it will pass through or be absorbed. However, the photonic crystal made of wing scales will provide a specific band gap in which incident light will not propagate but be reflected. The wavelength of the reflective light determines the color of the wing scales as human eyes see color via reflected light.

## 4. Discussions

### 4.1. Modeling and Calculation

As the two sections of *Papilio maackii* met the requirements of forming photonic crystals [24], the microstructures of these sections were simplified to 3D periodic models. Then, the plane-wave expansion method was applied to calculate the band gap of these photonic crystals.

The microstructures of the two sections of *Papilio maackii* were simplified to hexagonal photonic crystals, as shown in Figure 6a and Figure 7a. These photonic crystals were formed by chitin (frame) and air (voids), and the lattice constants are marked in the figures.

The E-polarization photonic bands of two sections of *Papilio maackii* are shown in Figure 6b and Figure 7b. It can be seen that both have partial photonic band gaps in each direction, as marked by the shaded areas. The presence of these areas means that light cannot propagate within a narrow range of wavelengths in the structure [8].

The band gaps of the two sections are summarized in Table 2. For each section, the band gaps have different, specific positions and various widths. This may explain why these sections show diverse colors.

### 4.2. Analysis

The frequency of the band gap can be correlated with the peak position of the reflective spectrum. The average normalized frequencies of band gaps at three directions are shown in Figure 8. Additionally, the peak positions of the reflective spectra (incident angle is 90°) of the two sections are displayed. The photonic band gap means there is a narrow range of wavelengths in which light cannot propagate but can only be reflected. As wavelength equals velocity/frequency, the frequency of the band gap is decided and inversely proportional to the wavelength of the reflective spectrum. The frequency of band gaps we obtained is eigenfrequency, which has a fixed relationship with frequency; thus, the eigenfrequency is also inversely proportional to the wavelength.

As can be seen from Figure 8, the normalized frequency of the band gap in Section 1 is lower than that in Section 2, which means the forbidden light has a longer wavelength. This light reflects and appears as a color to the naked eye. Therefore, the peak position of the reflective spectrum of Section 1 is further along than Section 2. This can also explain why Section 1 appears green while Section 2 appears blue in Figure 2 and Figure 3, owing to the wavelength of green light being higher than that of blue light.

## 5. Conclusions

*Papilio maackii* is a common species of butterfly living in temperate zones. Two sections of butterfly scales with different colors were analyzed in this paper. The microstructures were observed by a FESEM, and 3D periodic structures were acquired. The XPS results revealed that the chemical compositions of the two parts are absolutely the same. Reflective spectrums of scales were measured using UV-vis-NIR microspectrophotometers under incident light from different angles, and afterwards, the scales were immersed in ethanol. The results confirmed that the color of the two sections of *Papilio maackii* were decided by photonic crystals as their reflective spectra were influenced by a changing incident angle and interstitial substance.

We have proposed a 3D photonic crystal model to simplify scales of *Papilio maackii*, and the plane-wave expansion method was applied to calculate band gaps of the two sections. The relationship between the calculated band gap and practical reflective spectrum was established, and the normalized frequency of the band gap was inversely proportional to the peak wavelength of the reflective spectrum.

A full understanding of structural color in living nature can help us introduce the mechanism into colorful textiles, and this may revolutionize eco-dyeing technologies in textile industry, thus lowering the huge burden that the dyeing process imparts on the environment.

## Figures and Tables

**Figure 1 materials-15-03334-f001:**
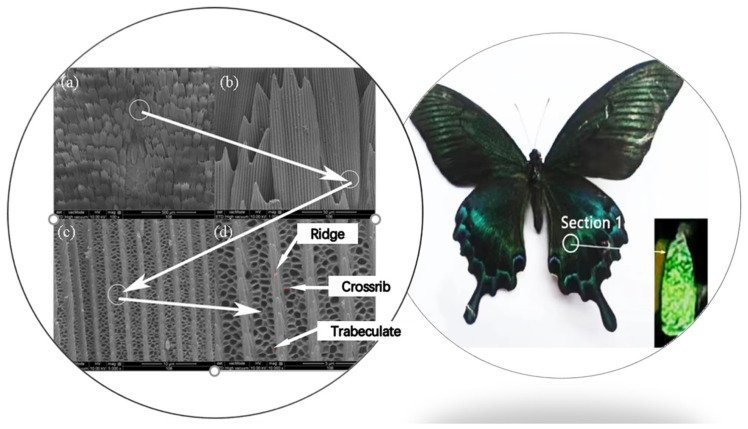
Images of Section 1 of *Papilio maackii* under different magnifications of (**a**) ×100, (**b**) ×1000, (**c**) ×5000, and (**d**) ×10,000.

**Figure 2 materials-15-03334-f002:**
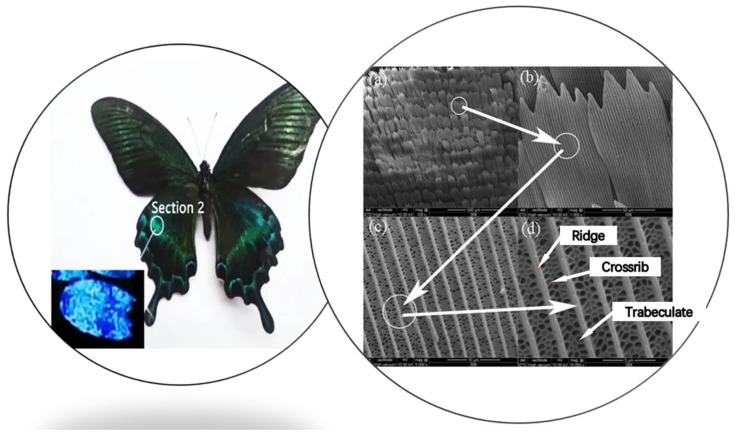
Images of Section 2 of *Papilio maackii* under different magnifications of (**a**) ×100, (**b**) ×1000, (**c**) ×5000, and (**d**) ×10,000.

**Figure 3 materials-15-03334-f003:**
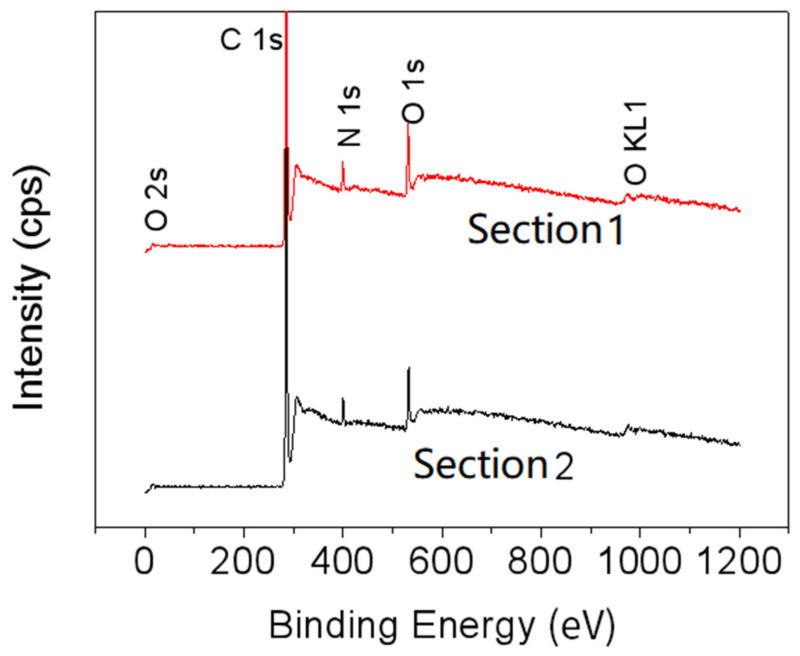
XPS spectra of wing scales of *Papilio maackii*.

**Figure 4 materials-15-03334-f004:**
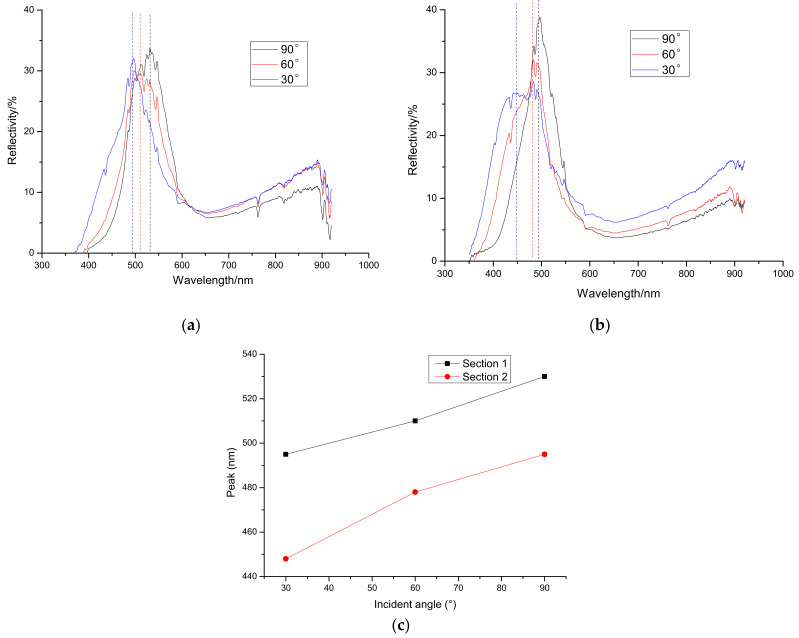
Reflective spectrum of incident light from different angles of (**a**) Section 1 and (**b**) Section 2 of *Papilio maackii*; (**c**) the relationship between the peak of the reflective spectrum and the incident angle.

**Figure 5 materials-15-03334-f005:**
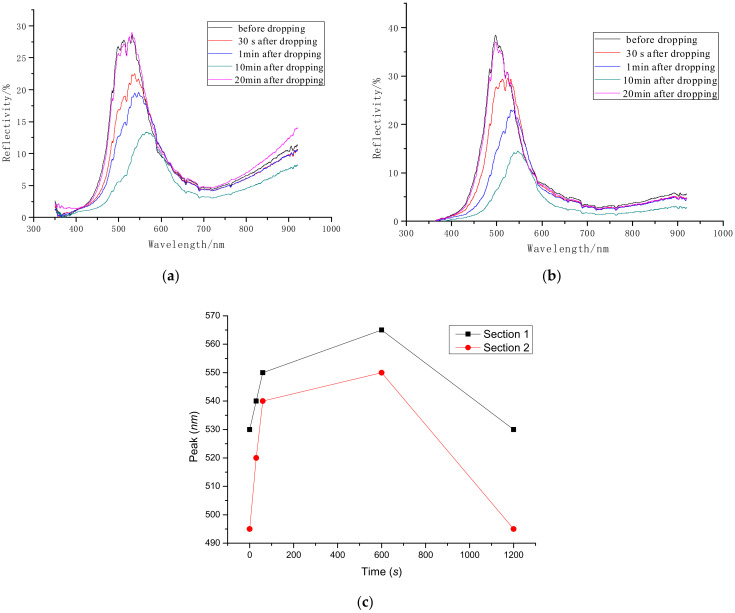
Reflective spectrum of scales after changing interstitial substance of (**a**) Section 1 and (**b**) Section 2 of *Papilio maackii*; (**c**) relationship between the time after dropping and the peak of the reflective spectrum.

**Figure 6 materials-15-03334-f006:**
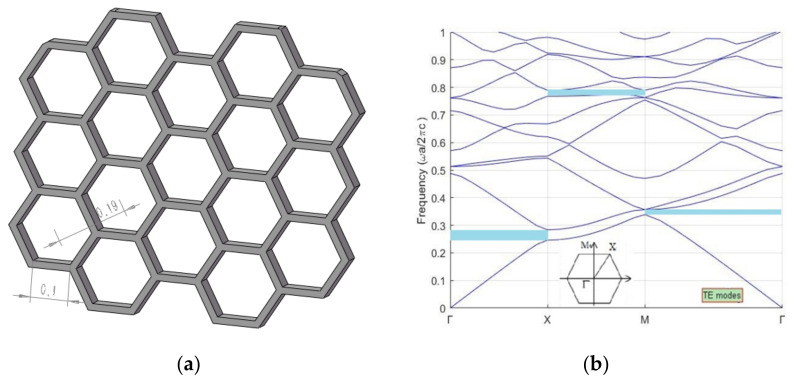
The microstructures of Section 1 of *Papilio maackii* form hexagonal photonic crystals: (**a**) a simplified model of scales; (**b**) E-polarization photonic band structure.

**Figure 7 materials-15-03334-f007:**
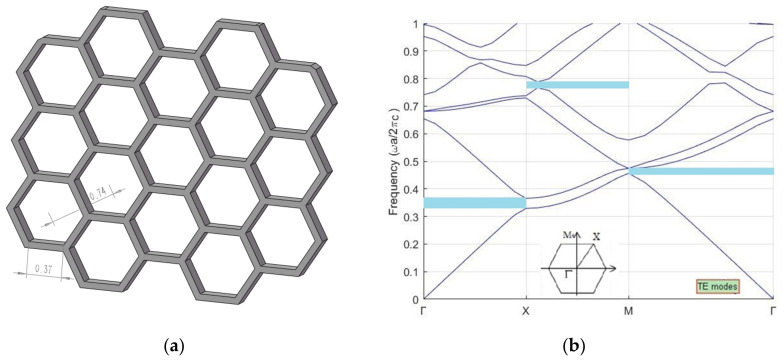
The microstructures of Section 2 of *Papilio maackii* form hexagonal photonic crystals: (**a**) a simplified model of scales; (**b**) E-polarization photonic band structure.

**Figure 8 materials-15-03334-f008:**
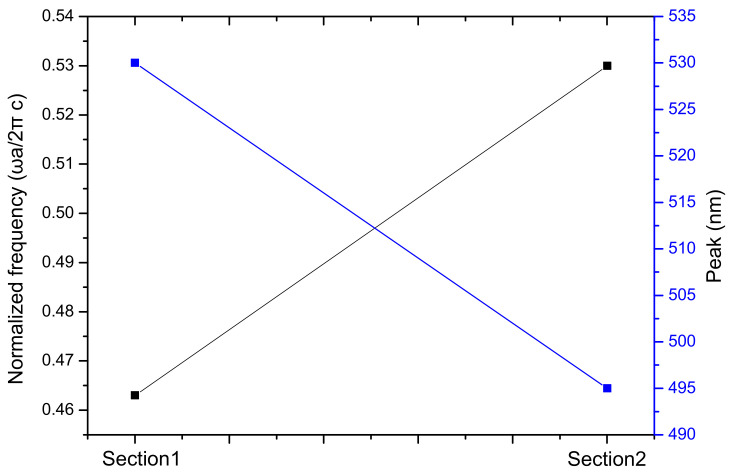
The relationship between the normalized frequency of the band gaps and the peak positions of the reflective spectrum.

**Table 1 materials-15-03334-t001:** Size comparison between two sections of *Papilio maackii*.

Size/μm	Length of Whole Scale	Width of Whole Scale	Spacing between Ridges	Thickness of Ridges	Spacing between Cross Ribs	Thickness of Cross Ribs
Section 1	132 ± 2	40 ± 2	1.75 ± 0.1	0.8 ± 0.1	0.74 ± 0.05	0.07 ± 0.01
Section 2	139 ± 3	57 ± 3	1.6 ± 0.1	0.55 ± 0.1	0.19 ± 0.02	0.08 ± 0.01

**Table 2 materials-15-03334-t002:** Normalized frequency (ωa/2πc) of the main band gaps in different sections.

	Section 1 of *Papilio maackii*	Section 2 of *Papilio maackii*
г-X	0.25–0.28	0.33–0.36
X-M	0.77–0.79	0.77–0.79
M-г	0.34–0.35	0.46–0.48

## Data Availability

Not applicable.

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
