# Peer review of "3D Modelling for Photonic Crystal Structure in Papilio maackii Wing Scales"

_materials, 2022, doi:10.3390/ma15093334_

Round 1

Reviewer 1 Report

This work studies the photonic properties of a structure taken from a butterfly. The results are self-consistent and the methods look solid. However, the writing (both English and organization) needs to be significantly improved. The current version looks more like a lab report rather a journal article. More crucially, I do not see any novelty and significance from the results presented by the authors. The fact that photonic crystal structures can exist in animals is quite well-known. From the results in this paper, I do not find anything special in this butterfly. Therefore, I cannot recommend its publication.

Author Response

Dear reviewer,

Thank you for your kind advice, we have revised the manuscript.

Hope for your new comments.

Yours,

Shu Yang

Reviewer 2 Report

General Comment:

The manuscript from Yang, Wang and Gao presents the study of Papilio maackii, a common species of butterfly living in China. The FESEM and the X-ray photoelectron spectroscopy were used to observe the microstructure of wing scales, and to clarify the chemical composition of butterfly wings. The optical property of wing scales was investigated by UV-vis-NIR micro spectrophotometer. The authors propose a simplified three-dimensional photonic crystal model according to the microstructure of wing scale, and they calculate the band gap. All characterizations are well conducted and are in adequacy with the journal “Polymers”. This study is is very descriptive, it would deserve to be supported by interpretations on the measured results (as example for reflective spectra). Some improvement could be realized before that this manuscript is published. I recommend publication after major revisions.

Comments and Minor points:

1- Material and methods: it should be good to give more details for FESEM as experimental conditions (electron tension, sample holder, secondary and backscatter electron,…). The SEM images are pretty nice, i just wonder why not show a magnification stronger than x10000 for section 1 and 2. on figures 2 and 3, it could be good to bring out the scale of the images, it is not easy to read. Looking at table 1, it is not always easy for non-specialists to identify each of the given parameters (length of whole scale, width of whole scale, spacing between ridges...) Wouldn't it be useful to add markers or identifications directly on the SEM images to specify what the authors are talking about in this table.

2- The figure 7 is rather blurred compared to figure 8, perhaps the authors could remedy this.

3- Part 4.1 modeling and Calculation; this part should be developed by the authors to give more explanation for the non-specialists. How are the values in Table 2 extracted from Figures 7 and 8? Also the use of specific terms in the table and in the text could be clearly explained (tau, X, M), and these terms should be defined.

4- The measurements made are of good quality but it would be useful to try to interpret more the results (UV-Visible, XPS, modeling and calculations). As an example, the reflective spectra measured are interesting, the authors should try to identify what are the electronic transitions associated with these observed bands leading to variation of the color. What explains the change in color or the band shift in reflectivity spectra?

5- The authors should read the text carefully to eliminate some typographical errors (missing spaces,...).

Author Response

Dear reviewer,

Thank you a lot for your kind advice, we have revised the manuscript according to you comments and made a  point-by-point response, please see the attachment. Hope for your new comments.

Yours,

Shu Yang

Reviewer 3 Report

Fig 1   - Bad quality scale photos - Include scale bars   Line 58   - How thick platinum?   Fig 2   - Where are the crossribs? The chitin walls making the hexagonal structure are not called crossribs in the literature.   Some voids seem to be closed, some seem to be open on the SEM image. Does it have a consequence?   Fig 6   You must incude a new figure, showing peak position and intensity as a function of time   Spectrum calculation and comparison of measured and calculated spectra are necessary.   Were there any difference between butterfly exemplars?   What is standard deviation of the measured structural parameters?   What is the biological significance of the results found?      

Author Response

(The authors gave the same response as above.)

Round 2

Reviewer 2 Report

The revisited manuscript from Yang, Wang and Gao presents the study of Papilio maackii, a common species of butterfly living in China. The authors have taken the time to answer the referees' questions/comments. Just a small point to put forward it is very difficult to know which images will remain and which will disappear from the final manuscript. I think the manuscript can now be published as is.

Author Response

Dear reviewer,

Thank you for your kind advice. Our manuscript has undergone English language editing by MDPI.

Yours,

Shu Yang

Reviewer 3 Report

The authors should fine some English proofreading service in order to improve the language and style. MDPI also offers such a service for an extra fee.

Author Response

(The authors gave the same response as above.)
